# NBT-Pluronic F-127 Hydrogels Printed on Flat Textiles as UV Radiation Sensors

**DOI:** 10.3390/ma14123435

**Published:** 2021-06-21

**Authors:** Elżbieta Sąsiadek, Malwina Jaszczak, Joanna Skwarek, Marek Kozicki

**Affiliations:** Department of Mechanical Engineering, Informatics and Chemistry of Polymer Materials, Faculty of Material Technologies and Textile Design, Lodz University of Technology, Żeromskiego 116, 90-924 Lodz, Poland; 219055@edu.p.lodz.pl (J.S.); marek.kozicki@p.lodz.pl (M.K.)

**Keywords:** UV radiation, radiochromic hydrogels, nitro blue tetrazolium chloride (NBT), Pluronic F-127, surface modification of textiles, UV radiation sensors

## Abstract

This work reports on the surface-modified woven fabrics for use as UV radiation sensors. The cotton and polyamide fabrics were printed with radiochromic hydrogels using a screen-printing method. The hydrogels used as a printing paste were composed of water, poly(ethylene oxide)-*block*-poly(propylene oxide)-*block*-poly(ethylene oxide) (Pluronic F-127) as a gel matrix and nitro blue tetrazolium chloride as a radiation-sensitive compound. The development of the hydrogels’ colour occurs after exposure to UV radiation and its intensity increases with increasing absorbed dose. The features of the NBT-Pluronic F-127 radiochromic hydrogels and the fabrics printed with the hydrogels were examined using UV-Vis and reflectance spectrophotometry as well as scanning electron microscopy (SEM). The effects of NBT concentration and UV radiation type (UVA, UVB, UVC) on dose responses of the hydrogels and printed fabrics were also examined. The results obtained reveal that the fabrics printed with NBT-Pluronic F-127 hydrogels can be potentially useful as UV radiation sensors.

## 1. Introduction

The negative effects of UV radiation on humans, in particular on the skin and eyes [1,2], were the driving force for research on systems for monitoring the amount of absorbed UV doses. Many sensors and dosimeters for UV monitoring have been elaborated, such as photodiodes and actinometers [3], liquid crystal mixtures, solutions of photoluminescent dyes [4,5] and inorganic materials [6,7,8,9,10]. These systems are most often placed at an appropriate distance from the source of UV radiation and then, after the agreed time, the changes to the dosimeters are read. Measurements performed in this way provide information on a radiation dose for a single measurement point (1D). Other systems for UV radiation measurements are dosimeters measuring with direct methods. An example is biological dosimeters based on polymers containing bacterial spores [11], vegetative bacteria [12] and DNA fragments of bacteriophages [13]. Additionally, chemical dosimeters or sensors are also used, which show greater accuracy and measurement stability compared to biological ones [14]. Examples of such systems are 2D film dosimeters and sensors in the form of polymer films containing radiation-sensitive compounds such as tetrazolium salts [15,16], triphenylmethane dyes [17] or polydiacetylenes [16]. Furthermore, other systems were proposed based on the surface modification of flat textiles with radiation-sensitive compounds by (i) textile padding with solutions containing selected tetrazolium salts [18,19,20,21] and (ii) screen-printing of textiles with a printing paste containing selected tetrazolium salts [21,22]. Moreover, the fibres doped with radiation-sensitive dyes from the group of tetrazolium salts and diacetylenes have also been manufactured [23,24]. These can be used to create 2D textile dosimeters for UV radiation measurements [24].

The aim of this work was a surface modification of flat textiles with hydrogels to obtain UV sensors for the potential use on the protective clothing for people exposed to this type of radiation. Following the former research [18,19,20,21,22], cotton and polyamide woven fabrics were selected for modification by a screen-printing method. However, instead of the typical printing paste containing thickening and crosslinking agents and a UV-sensitive compound, a radiochromic hydrogel as a printing paste was used. The hydrogel was composed of water, Pluronic F-127 gel matrix and nitro blue tetrazolium chloride (NBT) as a UV sensitive compound. The first reason for using it was to prevent the textile from stiffening after printing, which was guaranteed by the physical-chemical properties of Pluronic F-127. Moreover, many commercially used printing pastes do not form a transparent film on the textile (cloudiness and yellowing occur), which may cause readout errors. Pluronic F-127, on the other hand, is a copolymer composed of poly(propylene oxide) core and surrounding poly(ethylene oxide) blocks, which, if used as a gel matrix in radiochromic hydrogels, provides a high degree of transparency and colourlessness [25,26,27], which completely eliminates this problem. Furthermore, Pluronic F-127 is a nontoxic copolymer approved by the US Food and Drug Administration (FDA) [25], which is another reason for replacing the printing paste containing environmentally harmful chemical compounds. Finally, Pluronic F-127 shows a favourable phase behaviour—it is in the form of an aqueous solution or physical gel depending on the temperature and its concentration. For instance, a 25% Pluronic F-127 solution is in the form of a physical gel in the temperature range of approximately 19–85 °C [28]. At the temperature of transition into a physical gel (19 °C), it has a printing paste-like consistency.

So far, radiochromic hydrogels have been widely studied as dosimeters for high-resolution three-dimensional UV or ionising radiation dose distribution measurements. For this purpose, several dosimeter compositions have been tested, such as TTC-Pluronic F-127, LMG-Pluronic F-127 [25,26], NBT-Pluronic F-127 [29] and LCV-Pluronic F-127 [30,31]. In this work, we present another possibility of using such radiochromic hydrogels and, simultaneously, a new approach to the preparation of UV-sensitive textile-based sensors. For this aim, the NBT-Pluronic F-127 radiochromic hydrogel with a composition similar to that presented elsewhere [29] was selected.

## 2. Materials and Methods

### 2.1. Preparation of Samples

Hydrogel samples were composed of distilled water, poly(ethylene oxide)-*block*-poly(propylene oxide)-*block*-poly(ethylene oxide) (Pluronic F-127, Sigma-Aldrich, Saint Louis, MO, USA) matrix and nitro blue tetrazolium chloride (NBT, Roth, Karlsruhe, Germany,) as a radiation-sensitive compound. NBT was dissolved in distilled water at room temperature using a magnetic stirrer. Afterwards, the solution was placed in a refrigerator (5 °C) for 15 min and then gently mixed with cooled, 33% *w*/*w* Pluronic F-127 aqueous solution prepared 72 h earlier. The procedure of preparing Pluronic F-127 pre-solution was described elsewhere [25]. The final concentration of Pluronic F-127 in the hydrogel was equal to 25% *w*/*w*. After mixing, the solution was poured into poly(methyl methacrylate) (PMMA, Roth, Karlsruhe, Germany) cuvettes (4.5 cm^3^, 1 cm optical path) with four transparent optical faces, which were covered with matching polyethylene caps. Then, the samples were stored at room temperature (23 °C) and the sol-gel transition occurred after about 15 min. The gels were irradiated (Section 2.2) and measured with UV-Vis spectrophotometry (Section 2.4). The gel from the same batch was also used for printing of textiles (Section 2.3) followed by reflectance and scanning electron microscopy measurements (Section 2.5 and Section 2.7, respectively). The solutions/gels were protected from UV light by covering them with aluminium foil at every stage of their preparation and during storage.

### 2.2. Irradiation of Samples

Samples (gels in cuvettes and printed textiles) were irradiated in the UV-curing cabinets (UVP, Upland, Canada) at three wavelengths corresponding to UVA (8 W, type F8T5 Blacklight (range: 315–400 nm; a peak at 369 nm, Hitachi, Tokyo, Japan), UVB (8 W, type G8T5E (range: 280–360 nm; a peak at 306 nm, Sankyo Denki, Tokyo, Japan) and UVC (8 W, type G8T5, 253.7 nm, Sankyo Denki, Tokyo, Japan). Each cabinet was equipped with five UV lamps. A given UV dose (J/cm^2^) was delivered automatically using a built-in detector and the control system of the device. The samples were irradiated with UVA, UVB and UVC in the dose range of 0–3 J/cm^2^. During irradiation, the cuvettes with gels were uncovered.

### 2.3. Screen Printing

The polyamide and cotton woven fabric samples (A3 size) were screen-printed with the prepared hydrogels. For this purpose, a cotton fabric of a twill $\frac{3}{1}$S weave, a surface mass of 250 g/m^2^, warp 240/dm, weft 220/dm and thickness of 0.68 mm (Royal Ten Cate, Nijverdal, The Netherlands) and a polyamide fabric of a plain weave, a surface mass of 120 g/m^2^, warp 580/dm, weft 350/dm and thickness of 0.48 mm (Ortal S.A., Katowice, Poland) were used. A screen (EX 63-063/160 PW screen: 63 mesh/cm; thread diameter 63 µm; colour of mesh: white; tension: 18 N/cm; NBC, Tokyo, Japan; distributed by K+L Company, Lodz, Poland) was covered with a photosensitive emulsion (Fotocoat 1010; Forteco, Kwintsheul, The Netherlands) and dried in a dark place at 30 °C for 24 h. Then, a pattern (three 30 cm × 5 cm stripes, Figure 1) drawn and printed on a transparent foil (in black colour) was transferred to the screen, using a source of light (halogen lamp, Halogenfluter 500 W 930037; Düwi GmbH, Hamburg, Germany). The light emitted by the lamp caused polymerisation of the photosensitive emulsion in the places outside the pattern. The pattern areas were washed with cold water to remove non-polymerised emulsion. After drying the screen, the printing was carried out with the use of a squeegee. For printing, hydrogels were kept at 19 °C to keep the loose structure, resembling a printing paste. Then, printed fabrics were dried in a drier at 30 °C for 20 min and then left for 24 h at room temperature in a dark environment. Afterward, 4 cm × 4 cm samples were cut from the printed stripes, and these were irradiated with UV radiation.

### 2.4. Absorbance Measurements

A UV-Vis spectrophotometer (Jasco V-530, Tokyo, Japan, 190–1100 nm) was used for the measurements of NBT-Pluronic F-127 hydrogels in the PMMA cuvettes. The measurements were performed without an external access to UV light. The samples were measured 24 h after irradiation over the wavelength region of 450–700 nm. A background absorbance spectrum of the empty cuvette was subtracted from each gel absorbance spectra.

### 2.5. Reflectance Measurements

The reflectance spectra of cotton and polyamide fabrics printed with hydrogels were measured using Spectraflash light reflectance instrument (Spectraflash 300, D65/10; the measurement error is 0.1%, DataColor, Rotkreuz, Switzerland). The samples were measured 24 h after irradiation over the wavelength region of 400–700 nm. A UV light was cut off automatically by software in order not to irradiate the samples during measurements, which would falsify the results.

### 2.6. Evaluation of Printed Textile Surface Unevenness

To determine the unevenness of the print on the polyamide and cotton fabrics surface, the samples non-irradiated and irradiated with 0.5 and 2 J/cm^2^ doses of UVB light (selected from the linear dose range) were measured with an Epson Perfection V750 Pro scanner (Nagano, Japan; cold cathode fluorescent lamp; optical resolution Main 6.400 DPI × Sub 9.600 dpi; 48 bit/colour). The scanning was performed in the reflection mode, and the colour depth was 24-bit RGB. All samples (30 × 30 mm^2^) were scanned at a resolution of 300 dpi. Other scanning parameters like brightness, colour saturation correction, colour regulation and sharpness were switched off and were not taken into account in this study. The scans were used to calculate the profiles of the samples, which was conducted with the aid of a prepared script for reading RGB channels (RGBreader; Python Script with Python Imaging Library; DosLab [20]). Each sample was depicted using a three-colour RGB scale (red, green and blue). For the channel that showed the highest changes of values (from 0 to 255), further calculations were performed. After preliminary analysis of the examined samples, the green channel was chosen.

### 2.7. Scanning Electron Microscopy Measurements

The morphology of cotton and polyamide fabrics printed with NBT-Pluronic F-127 hydrogels was analysed with a TESCAN VEGA3–EasyProbe (TESCAN Brno, s.r.o., Brno, Czech Republic) scanning electron microscope equipped with a VEGATG software (high vacuum mode (SE); accelerating volt-age 20 kV). Before the measurements, all samples were coated with Au-Pd layers using a Cressington Sputter Coater 108 auto system (Cressington Scientific Instruments Ltd., Watford, UK).

## 3. Results and Discussion

### 3.1. Dose-Response of Hydrogels

The absorbance spectra of NBT-Pluronic F-127 hydrogels containing various concentrations of NBT (1, 2 and 5 g/dm^3^) irradiated with UVA, UVB and UVC radiation were registered. The color intensity of all samples increased with an increase in the absorbed dose, which is reflected in the absorbance spectra presented (Figure 2). This is related to the formation of the formazans (here: brownish-purple) upon radiation-induced reduction of NBT tetrazolium salt [32]. The formazans formed absorb visible light between 450 and 700 nm with a maximum at approximately 525 nm (Figure 2), which was confirmed by deconvolution of absorbance spectra for 3 J/cm^2^ dose of UVB. The most intense colour transformation was observed for samples irradiated with UVB radiation (Figure 2b,e,h) and the weakest for the samples exposed to UVC radiation (Figure 2c,f,i), which is reflected in the absorbance values. For instance, the absorbance at 525 nm of the samples containing the same amount of NBT and irradiated with 0.5 J/cm^2^ of UVB radiation is 20% and 121% higher than for samples irradiated with the same dose of UVA and UVC radiation, respectively (Figure 2a–c).

Based on the absorption spectra presented in Figure 2, the absorbance relations on absorbed dose for NBT-Pluronic F-127 gels at the maximum absorbance wavelengths were prepared (Figure 3). On their basis, the main features of the gels such as dose sensitivity, intercept, threshold dose, linear dose range and dynamic dose range were extracted and are shown in Table 1. The gels were the most sensitive to UVB radiation, regardless of the NBT concentration used. This was expressed as the highest dose-sensitivity values corresponding to the slope of the linear relations (Figure 3, Table 1). It is also visible in the photos showing the most intense change in colour of samples irradiated with UVB radiation (Figure 3). The weakest response was registered for the samples irradiated with UVC radiation. In the presented photographs (Figure 3), the changes in colour with increased absorbed dose for hydrogels irradiated with this type of radiation are not visible to the naked eye. However, these were recorded with a UV-Vis spectrophotometer, and the results are shown as changes in absorbance in Figure 2c,f,i. It should be noted that the part of UVC radiation may have been cut by the PMAA, although the cuvettes were opened during irradiation. To assess the impact of using PMMA cuvettes on the obtained results, the hydrogels in the PMMA and quartz cuvettes were irradiated with UVC radiation. The changes in colour were observed to the naked eye for the highest doses applied (above 0.5 J/cm^2^) for the samples in quartz cuvettes, whereas no colour formation in the irradiated gels was seen for the PMMA cuvettes. It was registered that there was an increase of absorbance at 525 nm by 6% and 43% for the samples in the quartz cuvettes irradiated with 0.5 J/cm^2^ and 0.7 J/cm^2^, respectively, compared to the samples irradiated in PMMA cuvettes.

The radiochromic hydrogels containing different concentrations of NBT and irradiated with different types of UV radiation show a wide linear dose range from less than 0.2 J/cm^2^ to at least 1.5 J/cm^2^. The dynamic dose range was observed in the entire studied range (from 0.01 to 3 J/cm^2^). The threshold dose for all gels studied was equal to 0.01 J/cm^2^, which indicates that hydrogels react to very low doses of UV radiation (Figure 3).

The dose responses of hydrogels with various concentrations of NBT were additionally compared (Figure 4). The samples were irradiated with UVB radiation, because all hydrogels reacted most strongly to this type of radiation, regardless of the amount of NBT in the composition (Figure 2 and Figure 3). The calculated parameters show that the dose sensitivity of hydrogels increases with an increase in NBT concentration and it is equal to 0.0691, 0.0719 and 0.0930 cm^2^/J for hydrogels with 1, 2 and 5 g/dm^3^ of NBT, respectively.

### 3.2. Dose–Response of Printed Textiles

The polyamide samples were printed with NBT-Pluronic F-127 hydrogels containing various concentrations of NBT (1, 2 and 5 g/dm^3^). After printing, the amount of paste applied to the fabrics was 30% by weight compared to the unprinted sample. We suspect that the interaction between NBT and cotton or polyamide fibers is related to the electro-chemical properties of the fibres and NBT. Electrokinetic potential measurements showed that NBT is of positive character in solution [22], whereas the fibres are negative [33,34]. Thus, NBT should be attracted by the fibres. This, in turn, should ensure adequate durability of the modified textiles during mechanical deformation. However, analysis of the impact of the mechanical deformations of textiles on the functionality deposited and dose–response were not considered in this work. In the next step, all the samples were irradiated with UVA, UVB and UVC radiation. The same effects of colour change as for the hydrogel samples were observed (Table 2). The samples were the most sensitive to UVB radiation, and the dose sensitivity increased with an increase in NBT concentration. However, for fabrics, the more intense change in colour with increasing absorbed dose was registered compared to hydrogels for all concentrations of NBT examined. Hence, the hydrogels with the lowest concentration of NBT (1 g/dm^3^) were used as a printing paste in further research.

To assess the effect of the textile substrate type on dose response, the polyamide and cotton samples were printed with NBT-Pluronic F-127 hydrogel containing 1g/dm^3^ NBT and were irradiated with UVB radiation. Then, the reflectance of light from the samples was measured. The results obtained are shown in Figure 5 and are expressed as the reflectance at 530 nm versus absorbed dose relations. The reflectance at 530 nm for cotton fabric printed with hydrogel decreased with an increase in absorbed dose in the entire studied range (0–1.5 J/cm^2^). For polyamide fabric printed with hydrogels, an increase in the light reflectance at 530 nm was observed for the highest doses of UVB (Figure 5) despite the visible increase in the colour intensity of the samples with an increase in the absorbed dose. This deviation is related to the unevenness of the print on the polyamide fabric surface. The hydrogel consists mainly of water, and the polyamide is hydrophobic and has low water absorption ability equal to 4–4.5% [35]. For this reason, the printed sample probably dried unevenly, resulting in its wrinkling. Due to the wrinkling of the material, the NBT particles could also be concentrated in the places of the material wrinkled and form locally the NBT formazan aggregates after irradiation, visible as areas of more intense colour. It was especially visible for samples irradiated with high doses of UV radiation in the range of 1–3 J/cm^2^. In the former studies [18,19,21], various attempts were made to obtain the colour uniformity of the modified polyamide woven fabrics. For this purpose, appropriate drying methods were used [18,22], and an additional surface modification using latex and gelatine was made [19,21]. Such approaches may be implemented also for polyamide fabrics printed with hydrogels, and this is the subject for further research.

Following the results presented in Figure 5a, the main features related to the dose responses of polyamide and cotton fabrics printed with NBT-Pluronic F-127 hydrogel were extracted and presented in Table 3. The dose sensitivity is more than two times higher for cotton than polyamide. Moreover, for the cotton–hydrogel system, the dynamic dose range is wider and the threshold dose is lower. The linear dose range is, on the other hand, much wider for the polyamide-hydrogel system. The stability over time after preparation of the printed cotton and polyamide samples can be described by the exponential decay curve according to the equation y = y_0_ + A$e^{-x/t}$ (Figure 5b). The stability of the printed and non-irradiated cotton and polyamide samples is similar. Five days after the preparation, the decrease in reflectance was approximately 16% (15.6% and 15.9%, respectively). Much greater differences are observed in the stability of printed samples and samples exposed to UVB radiation of 0.05 J/cm^2^. The printed cotton sample proved to be more unstable in the first days after irradiation. For this sample, the decrease in reflectance (at 530 nm) 5 days after irradiation was 26%, which is 13% greater than the decrease in reflectance of the printed polyamide fabric. However, in the following days, the sample stabilizes, while the reflectance of the polyamide sample still slightly decreases (by 3% from 5 to 7 days after irradiation).

A comparison of the colour change of samples with increasing absorbed dose expressed in CIELab system is presented in Table 4. The higher the absorbed dose, the more intense the colour change observed, for both polyamide and cotton samples as evidenced by the decreasing L-coordinate value with an increase in the absorbed dose (Table 4). It should be mentioned that textile raw materials are resistant to doses of UV radiation up to 20 J/cm^2^ [36]. The obtained photographs and the results of the analysis of the colour change in the CIELab system indicate that the intensity and shade of the obtained colour are influenced by the type of textile substrate used. The photographs show evenly printed cotton samples and unevenness on the printed polyamide samples due to the reasons described above. To produce a UV sensor, the cotton fabric should be used (in the currently elaborated manufacturing process) for printing to obtain uniformly distributed hydrogel.

### 3.3. The Unevenness Analysis of Fabrics Surface

To assess the unevenness of the print on the polyamide and cotton fabrics surface, the non-printed samples as well as the samples printed with NBT-Pluronic F-127 hydrogel and irradiated with 2 J/cm^2^ of UVB radiation were processed using RGBreader script described in Section 2.6 (Figure 6 and Figure 7). After scanning, each sample was saved as a bmp graphic file. In the next step, the RGBreader script was used, and each sample was described with three RGB channels. Based on preliminary tests, it was determined that the green RGB channel should be used for further analysis of the unevenness of the printed samples. The analysis of RGB channels for non-printed cotton and polyamide fabrics is presented in Figure 6.

The analysis was similarly performed for the samples irradiated with the dose of 2 J/cm^2^ UVB. In the case of the polyamide sample, the non-uniform colour in the green RGB channel is clearly visible (Figure 7a). This can result from the crease of the substrate and the formation of non-uniform printed areas due to the water repellency of the fabric. The results obtained for the cotton fabric indicate a higher uniformity of colour; however, the non-uniform weave of the fabric is also seen (Figure 7b). Although the weave of the fabric is a very important factor affecting the unevenness of the printed surface [20], the textile’s raw material is decisive in this case. Due to the technological and chemical parameters, the cotton fabric absorbed more NBT-Pluronic F-127 hydrogel solution into the structure. Moreover, the hydrophilic nature of cotton allowed for even distribution of the gel, which prevented the local increase in NBT concentration.

In the next stage, the profiles of green RGB channels across the samples were determined for polyamide and cotton samples before and after irradiation with UVB (Figure 8). The obtained results indicated that the polyamide fabric weave generates less disturbance of the printed surface unevenness (Figure 8a) compared to the cotton fabric weave (Figure 8b). The unevenness of the polyamide fabric also increases with the absorbed dose of UVB radiation. However, when analyzing the shape of the profiles, it can be seen that the uniformity of printing on the surface of cotton fabric is better than that for polyamide (Figure 8c). It is also seen that the non-printed cotton fabric is more uniform and creaseless.

### 3.4. Morphology of Textiles

The morphology of polyamide and cotton fabrics, non-printed and printed with NBT-Pluronic F-127 hydrogel containing 1 g/dm^3^ NBT was analysed and presented in Figure 9. The non-printed cotton fabric is much more contaminated (Figure 9a) than the polyamide fabric (Figure 9b). Moreover, it was visualized, that printing the samples with hydrogel changes their morphology. The hydrogel forms a film on the fibres surface and also penetrates visibly into the structure of the textiles (Figure 9c,d).

### 3.5. Proposition of Application

An example application of the elaborated method of radiochromic gel dosimeter deposition on textile materials is presented below. A warning pattern was designed (Figure 10a) to be printed on the textile with a radiochromic hydrogel, which may be used as a disposable element of protective clothing for workers exposed to UV radiation (Figure 10e). This pattern was printed on the cotton fabric using NBT-Pluronic F-127 hydrogel containing 1 g/dm^3^ NBT as a printing paste. Then, the change in colour was observed under exposure to UVB light (Figure 10b–d).

The developed systems can be used in various ways: for precise measurements of radiation doses if the system is modified to acquire higher stability, as well as for daily-use sensors, which provide information about overdose without giving an exact value of UV radiation. It should be noted that the unevenness of the print on the polyamide had an influence on the obtained results. Furthermore, the results indicate that the manufactured system can be used for a realistic exposure level of UV radiation measurements and SED (Standard Erythemal Dose) and MED (Minimum Erythemal Dose) ratios. For instance, SED and MED are 10 mJ/cm^2^ and 25 mJ/cm^2^, respectively, in Barcelona (Spain) [36], which is within the linear range of the measured UV doses proposed in this work’s systems.

The measurements of the absorbed dose of UV radiation by the proposed system can be performed as follows:a comparison of the pattern’s colour on the protective clothing with the prepared standard sample irradiated with a dose of UV radiation, which poses a threat to human health;a comparison of the pattern’s colour on the protective clothing with the prepared standard samples irradiated with different doses of UV radiation from the linear dose range of the proposed system;the reflectance of light measurements of the pattern on the protective clothing after a defined period of time and a readout of the absorbed dose from the calibration relation of the proposed system.

## 4. Conclusions

In this work, a proposition of a new UV sensor was shown, resulting from the surface modification of the flat woven fabrics using the screen printing method. The printing of textiles was presented using a novel printing paste formulation composed of water, Pluronic F-127 (gel matrix, printing paste base) and NBT (a UV-sensitive compound). The dose-response of hydrogels and printed fabrics (polyamide and cotton) was examined depending on the NBT concentration and UV radiation type. The samples change colour from a pale-yellow to brownish-purple under exposure to UV radiation and the colour intensity increases with increasing absorbed dose. Both hydrogels and printed fabrics are the most sensitive to UVB radiation, and their dose sensitivity increases with an increase in NBT concentration. More favorable results were obtained by the printing of cotton than polyamide. The sensor proposed in this work can be used as an element of protective clothing for workers exposed to UV radiation. The manufacturing method is simple, cheap and fast. A hydrogel used as a printing paste composed of more than 99% water and biocompatible Pluronic F-127, which, unlike the commercial printing pastes, is safe for an environment and as a print on clothing, which is in contact with the skin.

## Figures and Tables

**Figure 1 materials-14-03435-f001:**
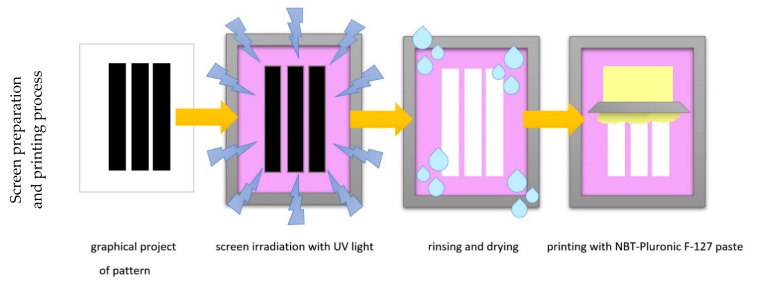
Schematic illustration of the screen-printing and irradiation process of flat woven fabric.

**Figure 2 materials-14-03435-f002:**
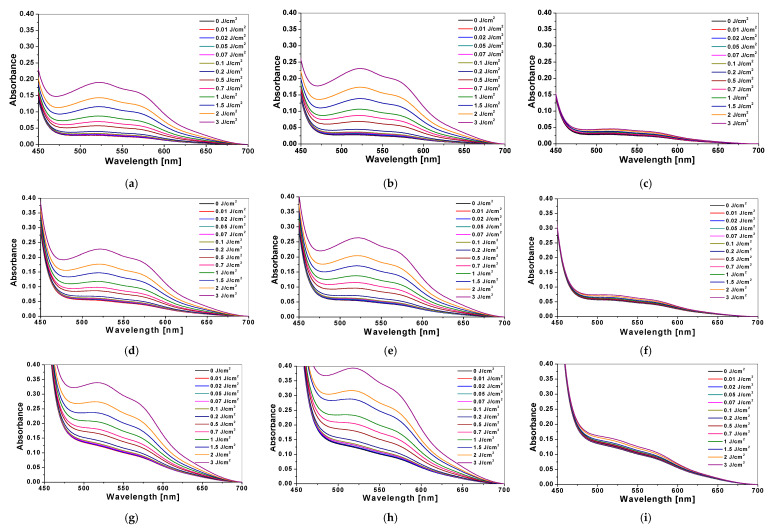
The absorbance spectra of NBT-Pluronic F-127 hydrogels with 1 g/dm^3^ (**a**–**c**), 2 g/dm^3^ (**d**–**f**) and 5 g/dm^3^ (**g**–**i**) NBT irradiated with UVA (**a**,**d**,**g**), UVB (**b**,**e**,**h**) and UVC (**c**,**f**,**i**) radiation in the dose range of 0–3 J/cm^2^.

**Figure 3 materials-14-03435-f003:**
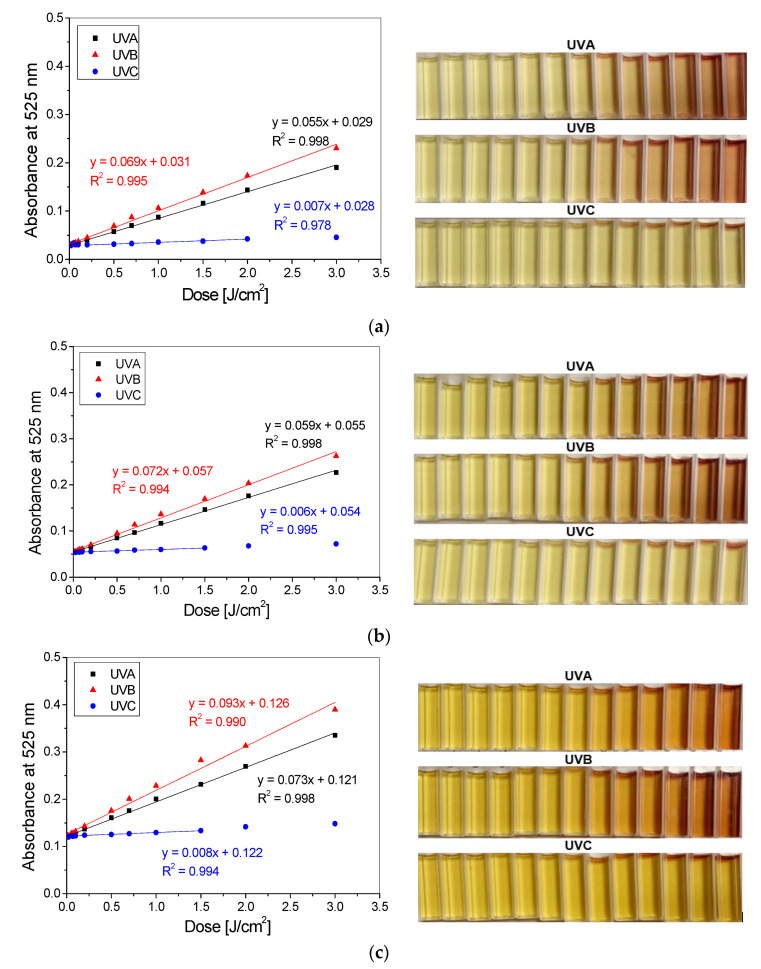
The dose responses of NBT-Pluronic F-127 hydrogels containing 1 g/dm^3^ (**a**), 2 g/dm^3^ (**b**) and 5 g/dm^3^ (**c**) NBT irradiated with UVA, UVB and UVC radiation in the dose range of 0–3 J/cm^2^ with corresponding photographs illustrating the samples’ colour changes after irradiation (absorbed doses: 0, 0.01, 0.02, 0.05, 0.1, 0.2, 0.5, 0.7, 1, 1.5, 2 and 3 J/cm^2^, from left to right).

**Figure 4 materials-14-03435-f004:**
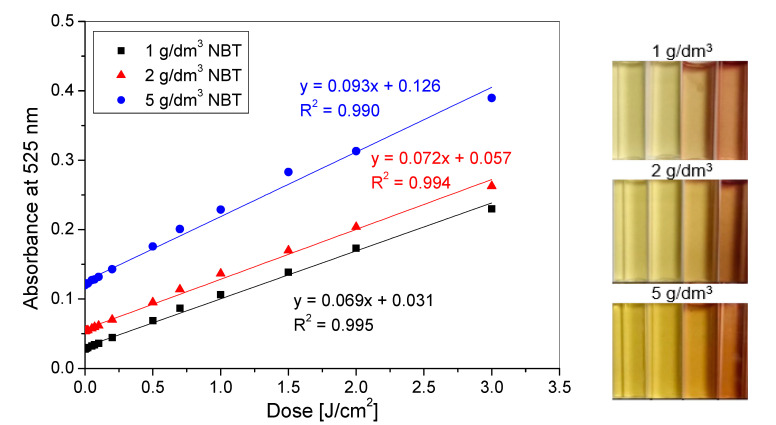
The dose responses of NBT-Pluronic F-127 hydrogels containing 1, 2 and 5 g/dm^3^ NBT irradiated with UVB radiation in the dose range of 0–3 J/cm^2^ with corresponding photographs illustrating the samples’ colour changes after irradiation with doses 0, 0.07, 0.7 and 3 J/cm^2^ (from **left** to **right**).

**Figure 5 materials-14-03435-f005:**
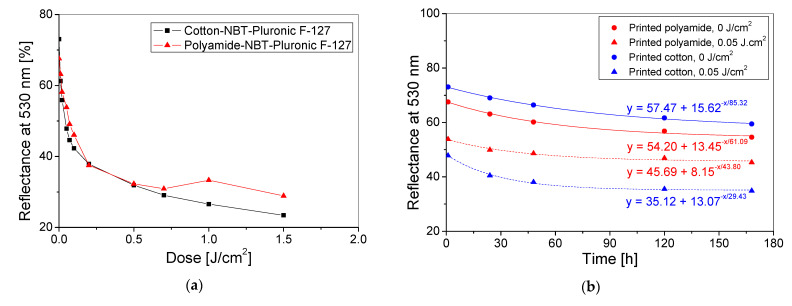
Dependence of the reflectance of light at 530 nm versus absorbed dose (**a**) and stability over time (**b**) for the cotton and polyamide fabrics printed with NBT-Pluronic F-127 hydrogel and irradiated with UVB radiation. The values of light reflectance for non-printed cotton and polyamide woven fabrics was: 79.60 and 72.60, respectively.

**Figure 6 materials-14-03435-f006:**
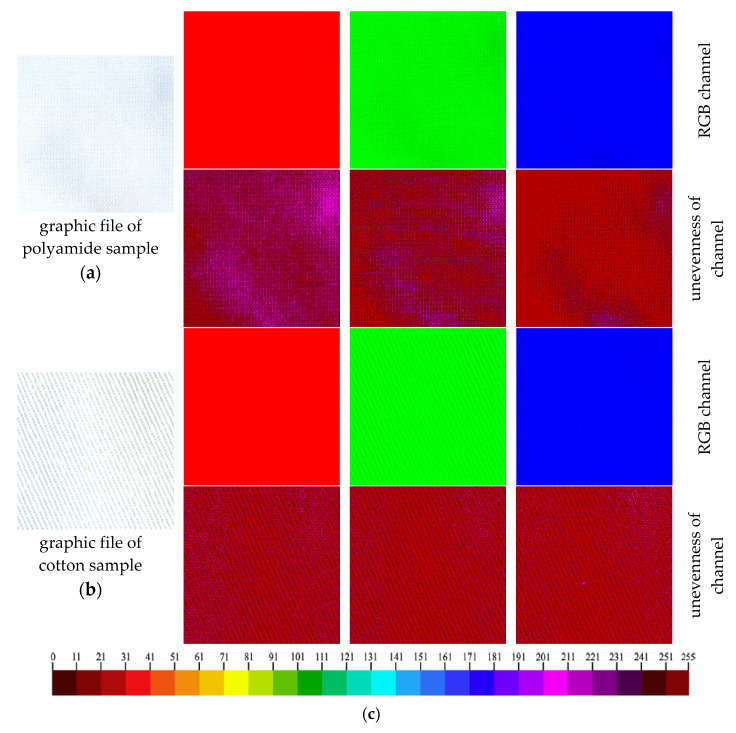
(**a**,**b**) Non-printed polyamide (**a**) and cotton (**b**) fabrics after RGBreader analysis; (**c**): The scale for the assessment of the unevenness of the RGB channel on the textiles’ surface (RGB values 0: ideal black; 255: ideal white) [20].

**Figure 7 materials-14-03435-f007:**
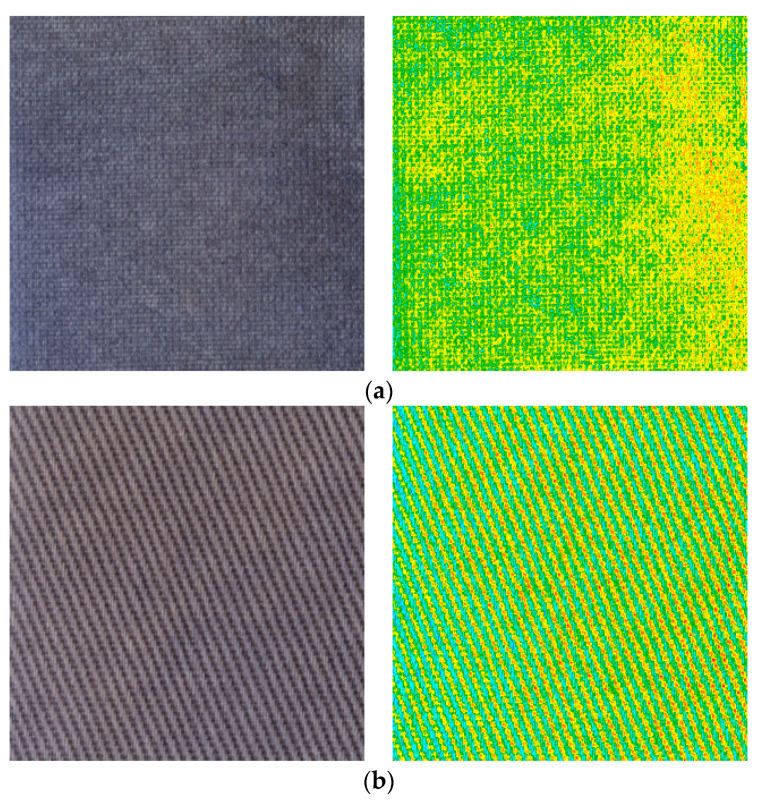
Green channels maps (second column) for polyamide (**a**) and cotton (**b**) fabrics printed with NBT-Pluronic F-127 hydrogel and irradiated with 2 J/cm^2^ of UVB radiation (first column).

**Figure 8 materials-14-03435-f008:**
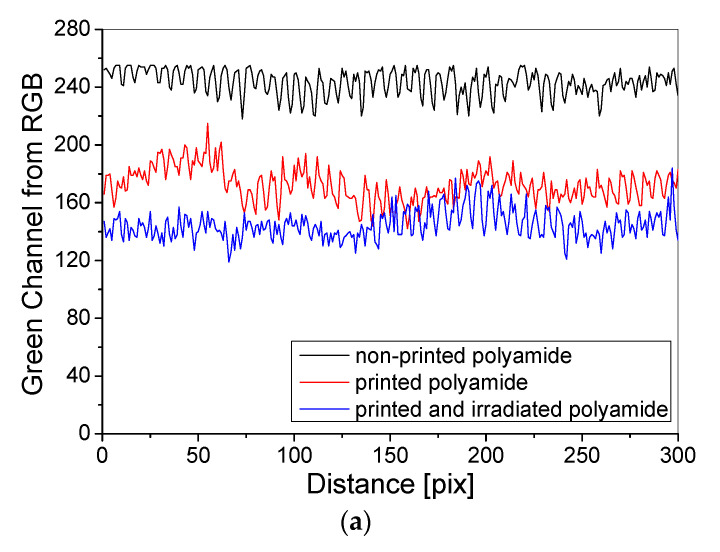
(**a**,**b**) Analysis of polyamide and cotton fabrics—the green RGB channel profiles (1 pix = 0.1 mm) for (i) non-printed, (ii) printed and non-irradiated and (iii) printed and irradiated with 0.5 J/cm^2^ of UVB irradiation polyamide (**a**) and cotton (**b**) fabrics; (**c**): Comparison of polyamide vs. cotton fabrics irradiated with 0.5 J/cm^2^ of UVB.

**Figure 9 materials-14-03435-f009:**
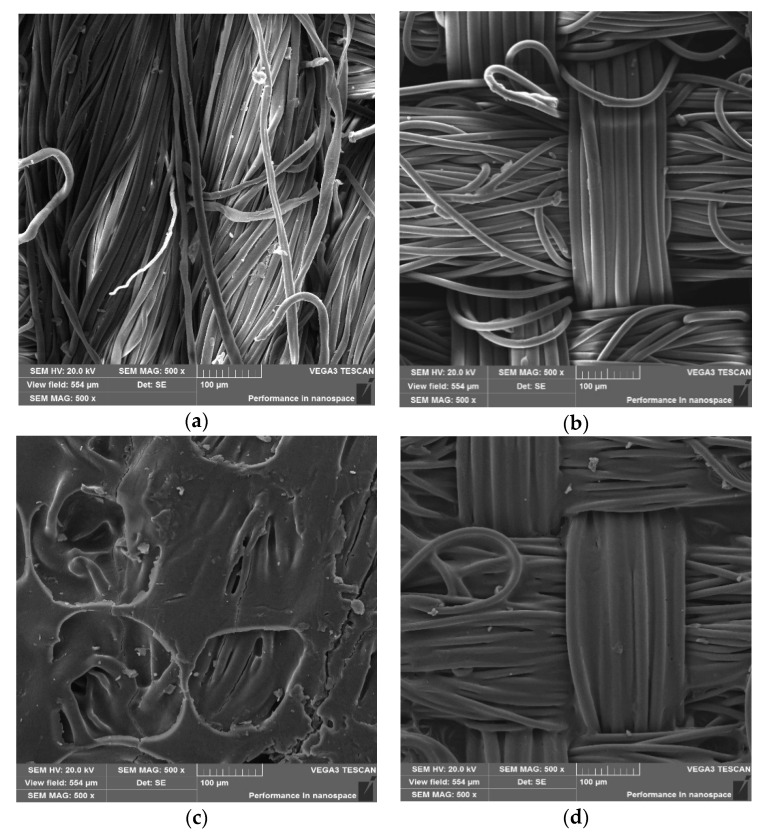
SEM images of the non-printed cotton fabric (**a**), non-printed polyamide fabric (**b**), cotton fabric printed with NBT-Pluronic F-127 hydrogel (**c**) and polyamide fabric printed with NBT-Pluronic F-127 hydrogel (**d**).

**Figure 10 materials-14-03435-f010:**
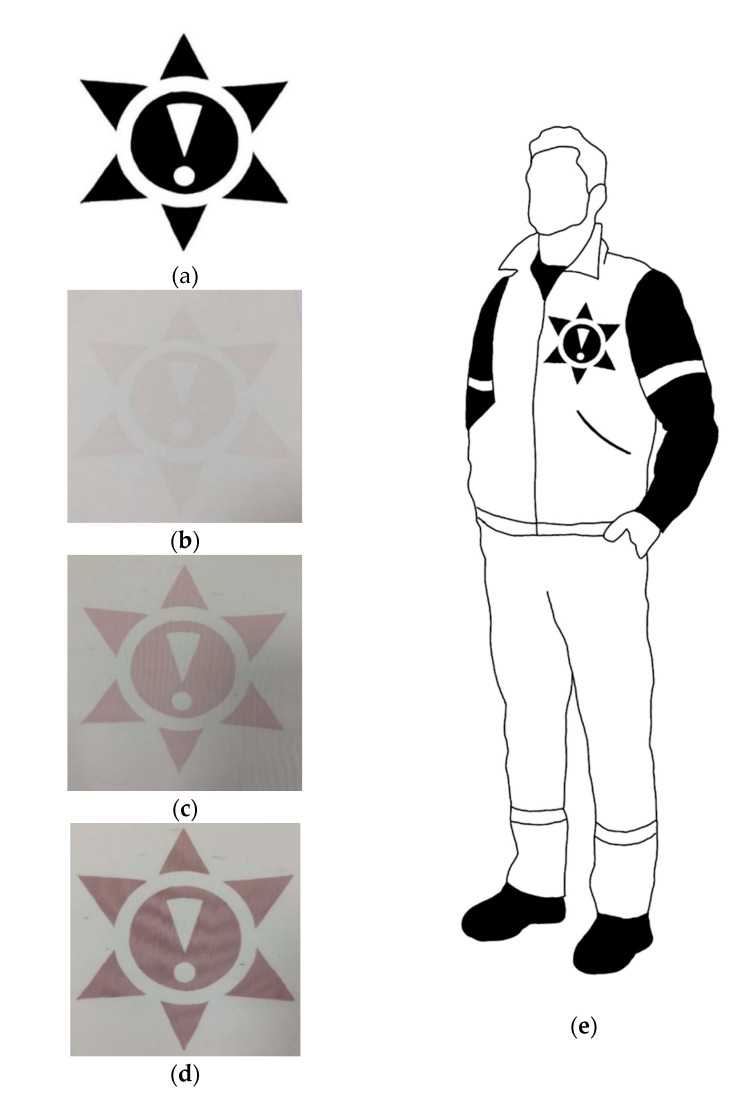
The pattern design (**a**); the pattern printed on cotton fabric with NBT-Pluronic F-127 hydrogel containing 1 g/dm^3^ NBT, non-irradiated (**b**) and irradiated with 0.05 J/cm^2^ (**c**) and 0.7 J/cm^2^ (**d**) of UVB radiation; an example of the location of the pattern on the protective clothing (**e**).

**Table 1 materials-14-03435-t001:** Basic characteristics of NBT-Pluronic F-127 gels with various concentrations of NBT and irradiated with UVA, UVB and UVC radiation.

The Range of UV Radiation	Concentration of NBT [g/dm^3^]	Dose Sensitivity [cm^2^/J]	Intercept [−]	Linear Dose Range [J/cm^2^]	Dynamic Dose Range [J/cm^2^]	Threshold Dose [J/cm^2^]	R^2^
UVA	1	0.0554 ± 0.0008	0.0291 ± 0.0009	0–3	0.01–3	0.01	0.998
UVB		0.0691 ± 0.0014	0.0311 ± 0.0016	0–3	0.01–3	0.01	0.995
UVC		0.0068 ± 0.0005	0.0280 ± 0.0005	0.2–2	0.01–3	0.01	0.978
UVA	2	0.0592 ± 0.0008	0.0545 ± 0.0009	0–3	0.01–3	0.01	0.998
UVB		0.0719 ± 0.0016	0.0566 ± 0.0018	0–3	0.01–3	0.01	0.994
UVC		0.0059 ± 0.0001	0.0541 ± 0.0001	0.01–1.5	0.01–3	0.01	0.995
UVA	5	0.0730 ± 0.0010	0.1213 ± 0.0012	0–3	0.01–3	0.01	0.998
UVB		0.0930 ± 0.0027	0.1259 ± 0.0031	0–3	0.01–3	0.01	0.990
UVC		0.0077 ± 0.0003	0.1219 ± 0.0002	0.1–1.5	0.01–3	0.01	0.984

**Table 2 materials-14-03435-t002:** The colour change of polyamide fabrics printed with NBT-Pluronic F-127 hydrogels containing various concentrations of NBT (1, 2 and 5 g/dm^3^), non-irradiated and irradiated with 0.05, 0.2 and 2 J/cm^2^ of UVA, UVB and UVC radiation.

**Printing Paste: NBT-Pluronic F-127 Hydrogel Containing 1 g/dm^3^ NBT** **Irradiation: Various Types of UV Radiation**					
	Absorbed dose	0 J/cm^2^	0.05 J/cm^2^	0.2 J/cm^2^	2 J/cm^2^
UV radiation					
UVA		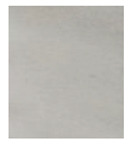	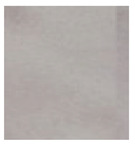	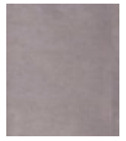	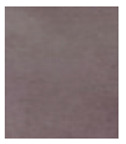
UVB		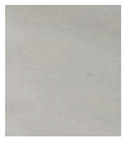	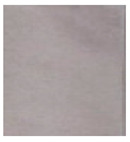	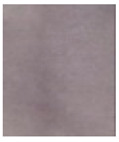	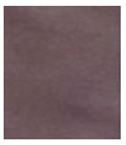
UVC		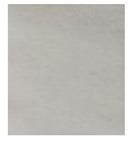	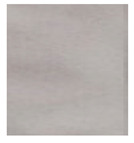	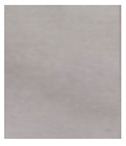	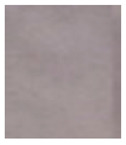
**Printing Paste: NBT-Pluronic F-127 Hydrogels Containing Various Concentrations of NBT** **Irradiation: UVB Radiation**					
	Absorbed dose	0 J/cm^2^	0.05 J/cm^2^	0.2 J/cm^2^	2 J/cm^2^
NBT concentration					
1 g/dm^3^		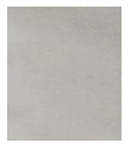	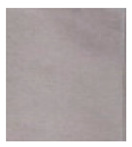	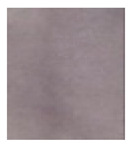	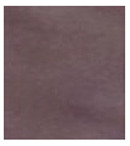
2 g/dm^3^		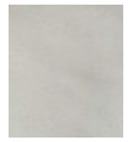	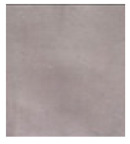	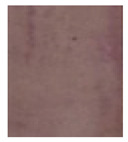	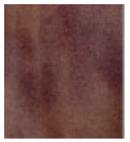
5 g/dm^3^		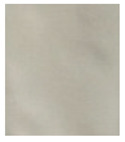	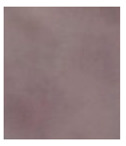	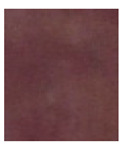	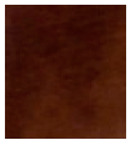

**Table 3 materials-14-03435-t003:** Basic characteristics of polyamide and cotton woven fabric printed with NBT-Pluronic F-127 hydrogels and irradiated with UVB radiation.

Woven Fabric	Dose Sensitivity [% × cm^2^/J]	Intercept [%]	Linear Dose Range [J/cm^2^]	Dynamic Dose Range [J/cm^2^]	Threshold Dose [J/cm^2^]	R^2^
Polyamide-NBT-Pluronic F-127	−113.88 ± 13.24	59.42 ± 1.42	0.02–0.2	0.02–1.5	0.02	0.966
Cotton-NBT-Pluronic F-127	−269.81 ± 36.24	62.51 ± 1.61	0.01–0.07	0.01–1.5	0.01	0.948

**Table 4 materials-14-03435-t004:** The colour change of polyamide and cotton fabrics printed with NBT-Pluronic F-127 hydrogel and irradiated with UVB radiation.

Absorbed Dose [J/cm^2^]	Polyamide Woven Fabric				Cotton Woven Fabric			
	Photographs	L	a	b	Photographs	L	a	b
non-printed	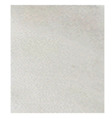	90.82	0.14	2.01	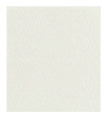	91.38	0.48	2.12
0	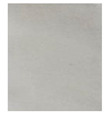	86.30	0.08	5.64	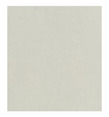	89.12	0.20	6.68
0.01	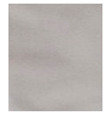	82.56	2.70	3.04	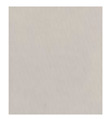	84.08	3.85	4.33
0.02	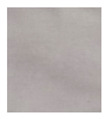	80.78	3.68	2.31	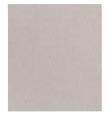	81.51	5.45	3.41
0.05	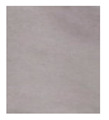	78.08	4.26	2.02	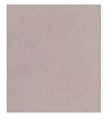	77.17	7.41	2.44
0.1	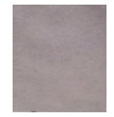	74.77	4.93	1.59	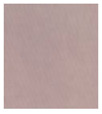	73.73	8.40	1.95
0.2	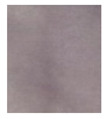	68.18	5.96	0.78	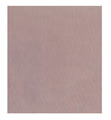	70.61	8.76	2.34
0.5	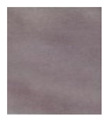	65.41	5.69	0.64	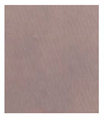	65.91	8.73	3.92
1	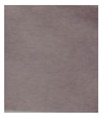	62.09	5.60	1.84	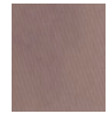	61.21	8.83	4.61
1.5	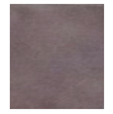	58.56	5.51	2.56	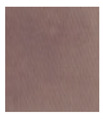	57.47	8.31	4.71

## Data Availability

Data available on request by contacting with the corresponding authors.

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
