# Peer review of "NBT-Pluronic F-127 Hydrogels Printed on Flat Textiles as UV Radiation Sensors"

_materials, 2021, doi:10.3390/ma14123435_

Round 1
Reviewer 1 Report
This manuscript by Sąsiadek et al. introduces a UV-sensitive hydrogel printed on a fabric as a UV sensor. Overall, this study introduces an interesting and useful application of hydrogel. However, a few things need to be addressed.
-Major comment-
- It is important that the hydrogel-printed fabric should have adequate mechanical strength, and able to mechanically deform without hydrogel detachment or damage. Measure the tensile mechanical properties of the fabric and hydrogel-printed fabric. Also, measure the UV-induced color change at various mechanical deformations.
- In addition to photographs of fabrics in Figure 2, it would be better to also include a schematic illustration of the screen-printing process.
- Figure 1 was previously used in another research article by the same author (J. Photochem. Photobiol. A: Chem, 2021, 405, 112930). Add the rights and permission statement in the figure caption or just simply remove it, as it is not necessary.
-Other comments-
- Provide the fabrication of cotton fabric and polyamide fabric used in this study. If they were purchased, provide the name of the vendor.
- There are no error bars in any of the graphs, so it is not possible to assess the statistical significance.
Reviewer 2 Report
The manuscript proposed a tetrazolium salt doped Pluronic F127 hydrogel printing technique on textiles. The idea of printing stimuli-responsive Pluronic F127 (with and without added species) on textile was explored previously. However, the data, particularly the interaction with UVs on the printed gel, presented in the manuscript is new and meaningful for this particular material and application. The manuscripts require significant adjustments before it is suitable to be published in Materials. The numbered comments are as follows:
- The manuscript completely disregards material chemical analysis. The reviewer suggests any spectroscopic analysis techniques which confirm the presence of Pluronic-F127, NBT, formazans (after irradiation), etc.
- Why the energy range of 0-3 J/cm3 was used? What were the realistic exposure level of UV radiation in the environment and OSHA's regulations?
- Why include Figure 1 at all? Were these devices custom-made? Do the numbers on the LCD have any meaning?
- Line 125, the reviewer assumes that the printed hydrogel was dried at 30oC for 20 minutes then at room temperature for 24 hours in a dark environment? The reviewer suggested modifications of the sentence.
- In section 2.6, the author mentioned that the photo was selected before being analyzed. What were the selection criteria? How was this selection process affect the data being presented?
- The author does not include any error bars in Figures 4, 5, 6. Please include error bars.
- The reviewer suggests adding non-coated fabrics as controls in Figure 6.
- To reduce confusion, the use of simply "cotton" and "polyamide" (i.e., Figure 6) in the place of hydrogel-coated cotton and polyamide should be reconsidered.
- Since all measurement was done after 24 hours, how does radiation exposure time affect the observation? Was the color develop in a linear, inverse exponential, or other forms of progression?
- In table 3 and 4. the reviewer strongly suggest the author include non-coated fabrics as controls. As UVs exposure is known to damage the fabric naturally and the sensor needed 24 hours of exposure to develop color intensity as reported, the effects of the amount of UV being exposed on the control fabrics are needed for completeness.
- The reviewer suggested adding deviation in the value reported in table 4 without deviation, the question of reproducibility is of concern.
- In table 3. please provide a rationale on the lack of deviation on dose sensitivity when compared to table 1.
- The conversion of NBT to formazan through UV irradiation is not reversible and will likely progress over time with regular exposure to low doses of UV. How will the author justify the accuracy of the measurement in practice?
- Why was the Green channel was selected for the unevenness analysis? Please provide a rationale or a reference.
- In the conclusion, Line 372-373, the author claimed a novelty on textile printing. However, the UVs responses results are new but that is not the case for the printing approach. Please provide a strong rationale for the claim.
Reviewer 3 Report
In this form, the article is not suitable for publication and needs a serious revision.
- In the introduction, the mention of “exploiting inorganic materials” looks pathetic and unworthy. Quoting a very old patent here that is not relevant here is puzzling. It should be noted here that, for example, a large number of materials showing outstanding photostimulated luminescence properties show corresponding dosimetric properties with a a very large dynamic range and very high sensitivity ( up to 10-12 J/cm2), see Fig. 3 in paper: Popov, A. I., & Plavina, I. (1995). Photostimulated emission of KBr—In previously exposed to UV- or X-radiation. Nuclear Instruments and Methods in Physics Research Section B: Beam Interactions with Materials and Atoms, 101(3), 252–254. https://doi.org/10.1016/0168-583X(95)00485-8
Apart from KBr:In, other similar inorganic crystalline materials were considered:
de Cárcer, I. A., Dántoni, H. L., Barboza-Flores, M., Correcher, V., & Jaque, F. (2009). KCl: Eu2+ as a solar UV-C radiation dosimeter. Optically stimulated luminescence and thermoluminescence analyses. Journal of Rare Earths, 27(4), 579-583.
Rivera, T., Azorı́n, J., Furetta, C., Falcony, C., Garcı́a, M., & Martı́nez, E. (2004). Green stimulated luminescence of ZrO2+ PTFE to UV radiation dosimetry. Nuclear Instruments and Methods in Physics Research Section B: Beam Interactions with Materials and Atoms, 213, 325-328.
Smetana, F., Hajek, M., Bergmann, R., Brusl, H., Fugger, M., Gratzl, W., ... & Vana, N. (2008). A portable multi-purpose OSL reader for UV dosimetry at workplaces. Radiation measurements, 43(2-6), 516-519.
Schuyt, J. J., & Williams, G. V. M. (2019). Optical properties of Mn2+ doped CsCdF3: A potential real-time and retrospective UV and X-ray dosimeter material. Journal of Applied Physics, 125(23), 233102.
and references therein.
- The spectra in Figure 3 should be decomposed into individual spectral components (in eV or cm-1) and only after that the appropropriate dose dependences should be analyzed for individual absorption bands. Otherwise, the analysis looks incorrect.
- Since the journal is called “Materials”, it would be useful in the conclusion to see a clearly formulated conclusion - what new data about the materials have been obtained and how they are compared with others, especially in terms of spectral sensitivity, dynamic range, fading, multiple use etc.
Round 2
Reviewer 1 Report
The revision has addressed the questions adequately. No further comments.
Reviewer 2 Report
The author has sufficiently addressed the reviewer's comments.
Reviewer 3 Report
The authors significantly improved the manuscript and answered the questions completely. The manuscript can be accepted